# Revitalizing CNN Attentions via Transformers in Self-Supervised Visual Representation Learning

**Chongjian Ge**[1]    **Youwei Liang**[2]    **Yibing Song**[2*]    **Jianbo Jiao**[3]    **Jue Wang**[2]    **Ping Luo**[1]

[1]The University of Hong Kong    [2]Tencent AI Lab    [3]University of Oxford

rhettgee@connect.hku.hk    liangyouwei1@gmail.com    yibingsong.cv@gmail.com

jianbo@robots.ox.ac.uk    arphid@gmail.com    pluo@cs.hku.hk

## Abstract

Studies on self-supervised visual representation learning (SSL) improve encoder backbones to discriminate training samples without labels. While CNN encoders via SSL achieve comparable recognition performance to those via supervised learning, their network attention is under-explored for further improvement. Motivated by the transformers that explore visual attention effectively in recognition scenarios, we propose a CNN Attention REvitalization (CARE) framework to train attentive CNN encoders guided by transformers in SSL. The proposed CARE framework consists of a CNN stream (C-stream) and a transformer stream (T-stream), where each stream contains two branches. C-stream follows an existing SSL framework with two CNN encoders, two projectors, and a predictor. T-stream contains two transformers, two projectors, and a predictor. T-stream connects to CNN encoders and is in parallel to the remaining C-Stream. During training, we perform SSL in both streams simultaneously and use the T-stream output to supervise C-stream. The features from CNN encoders are modulated in T-stream for visual attention enhancement and become suitable for the SSL scenario. We use these modulated features to supervise C-stream for learning attentive CNN encoders. To this end, we revitalize CNN attention by using transformers as guidance. Experiments on several standard visual recognition benchmarks, including image classification, object detection, and semantic segmentation, show that the proposed CARE framework improves CNN encoder backbones to the state-of-the-art performance.

## 1   Introduction

Learning visual features effectively has a profound influence on the recognition performance [5, 53]. Upon handling large-scale natural images, self-supervised visual representation learning benefits downstream recognition tasks via pretext feature training. Existing SSL methods typically leverage two branches to measure the similarity between different view representations derived from the same input image. By maximizing the similarity between the correlated views within one image (e.g., BYOL [24], SimSiam [14] and Barlow Twins [73]), or minimizing the similarity between views from different images (e.g., MoCo [27] and SimCLR [12]), these methods are shown to be effective towards learning self-supervised visual representations.

SSL evolves concurrently with the transformer. Debuting at natural language processing [62, 17], transformers have shown their advantages to process large-scale visual data since ViT [19]. The encoder-decoder architecture in this vision transformer consistently explores global attention without convolution. This architecture is shown effective for visual recognition with [7, 76, 54] or without CNN integration [38, 21]. Inspired by these achievements via supervised learning, studies [15, 10, 4, 71] arise recently to train transformers in a self-supervised manner. These methods maintain most of

---

*Y. Song is the corresponding author. The code is available at https://github.com/ChongjianGE/CARE

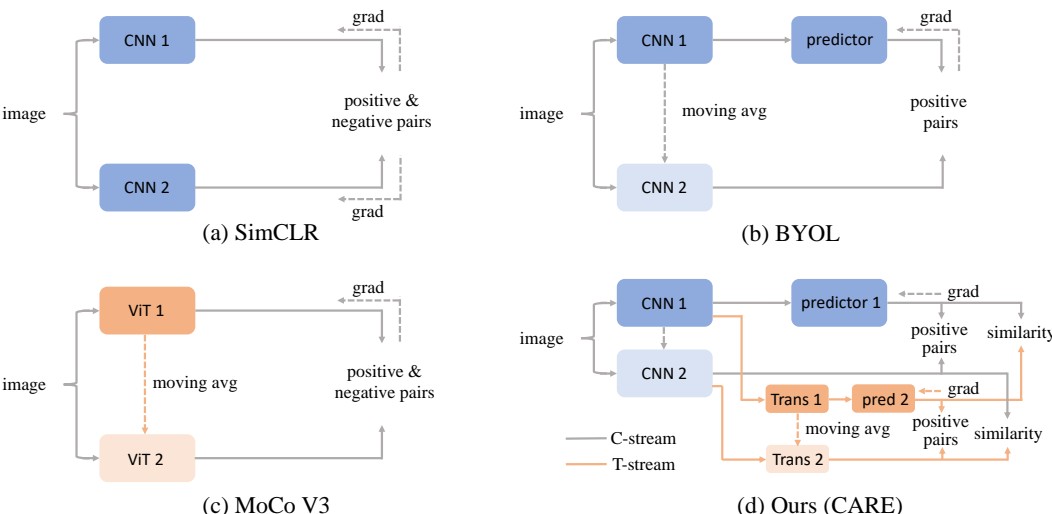

Figure 1: SSL framework overview. The solid lines indicate network pipeline, and the dash lines indicate network updates. MoCo V3 [15] explores visual attention by explicitly taking a vision transformer as encoder, while SimCLR [12] and BYOL [24] do not learn an attentive CNN encoder. Our CARE framework consists of a C-stream and a T-stream to explore visual attention in CNN encoders with transformer supervision. Note that only target CNN encoder (i.e., $CNN_1$) is preserved after pre-training for downstream evaluation. We do not show projectors in (b) and (d) for simplicity.

the SSL pipeline (i.e., encoder, projector, and predictor) utilized for training CNN encoders. Without significant framework alteration, original SSL methods for CNN encoders can be adapted to train transformer encoders and achieve favorable performance.

The success of using transformer encoders indicates that visual attention benefits encoder backbones in SSL. On the other hand, in supervised learning, CNN attention is usually developed via network supervision [47]. However, we observe that existing SSL methods do not incorporate visual attention within CNN encoders. This motivates us to explore CNN attention in SSL. We expect CNN encoders to maintain similar visual attention to transformers for recognition performance improvement with lower computational complexity and less memory consumption.

In this paper, we propose a CNN Attention REvitalization framework (CARE) to make CNN encoder attentive via transformer guidance. Fig. 1 (d) shows an intuitive illustration of CARE and compares it with other state-of-the-art SSL frameworks. There are two streams (i.e., C-stream and T-stream) in CARE where each stream contains two branches. C-stream is similar to existing SSL frameworks with two CNN encoders, two projectors, and one predictor. T-stream consists of two transformers, two projectors, and one predictor. T-stream takes CNN encoder features as input and improves feature attention via transformers. During the training process, we perform SSL in both streams simultaneously and use the T-stream output to supervise C-stream. The self-supervised learning in T-stream ensures attentive features produced by transformers are suitable for this SSL scenario. Meanwhile, we use the attention supervision on C-stream. This supervision enables both C-stream and T-stream to produce similar features. The feature representation of CNN encoders is improved by visual attention from transformers. As a result, the pre-trained CNN encoder produces attentive features, which benefits downstream recognition scenarios. Experiments on standard image classification, object detection, and semantic segmentation benchmarks show that the proposed CARE framework improves prevalent CNN encoder backbones to the state-of-the-art performance.

## 2    Related works

In the proposed CARE framework, we introduce transformers into self-supervised visual representation learning. In this section, we perform a literature survey on related works from the perspectives of visual representation learning as well as vision transformers.

## 2.1 Visual representation learning

There is an increasing need to learn good feature representations with unlabeled images. The general feature representation benefits downstream visual recognition scenarios. Existing visual representation learning methods can be mainly categorized as generative and discriminative methods. The generative methods typically use an auto-encoder for image reconstruction [63, 49], or model data and representation in a joint embedding space [18, 6]. The generative methods focus on image pixel-level details and are computationally intensive. Besides, further adaption is still required for downstream visual recognition scenarios.

The discriminative methods formulate visual representation learning as sample comparisons. Recently, contrastive learning is heavily investigated since its efficiency and superior performance. By creating different views from images, SSL obtains positive and negative sample pairs to constitute the learning process. Examples include memory bank [70], multi-view coding [58, 61], predictive coding [30, 60], pretext invariance [42], knowledge distillation [22, 10] and information maximization [32]. While negative pairs are introduced in MoCo [27] and SimCLR [12], studies (e.g., BYOL [24] and SimSiam [14]) show that using only positive pairs are effective. Also, clustering methods [8, 9] construct clusters for representation learning. The negative pairs are not introduced in these methods. Besides these discriminative methods focusing on image data, there are similar methods learning representations from either video data [66, 26, 33, 67, 45, 65, 64] or multi-modality data [1, 2, 3]. Different from these SSL methods, we use the transformer architectures to improve CNN encoders attention.

## 2.2 Vision transformers

Transformer is proposed in [62] where self-attention is shown effective for natural language processing. BERT [17] further boosts its performance via self-supervised training. The sequential modeling of transformer has activated a wide range of researches in natural language processing [57], speech processing [56], and computer vision [25]. In this section, we only survey transformer-related works from the computer vision perspective.

There are heavy researches on transformers in both visual recognition and generation. ViT [19] has shown that CNN is not a must in image classification. DETR [7], Deformable DETR [78] and RelationNet++ [16] indicate that transformers are able to detect objects with high precisions. SETR [76] brings transformers into semantic segmentation while VisTR [69] has shown transformers are able to perform video object segmentation. TrackFormer [40] introduces transformers into multiple object tracking. A general form of transformer is formulated in NLM [68] for video classification. Furthermore, transformers have been show effective in image generation [46] and image processing [11] scenarios. Examples include image super-resolution [72], video inpainting [74], and video captioning [77]. There are several emerging studies [15, 10, 4, 71] on how to use self-supervised learning to improve a transformer backbone. The learning paradigm for CNN encoders is adapted to the transformer without significant alteration. Different from existing methods that focus on learning a transformer encoder backbone with supervised or self-supervised learning. We explore how to use the transformers as guidance to enhance CNN visual attention. The pretrained CNN encoder benefits downstream recognition scenarios.

## 3   Proposed method

Our CARE framework consists of C-stream and T-stream. Fig. 2 shows an overview of the pipeline. We first illustrate the detailed structure of these streams. Then, we illustrate the network training process. The CNN encoder features are visualized as well for attention display.

## 3.1 CNN-stream (C-stream)

Our C-stream is similar to the existing SSL framework [24] where there are two CNN encoders, two projectors, and one predictor. The structures of the two encoders are the same, and the structures of the two projectors are the same. Given a training image $x$, we process it with a set of random augmentations to create two *different* augmented views. We feed these two views to C-stream and obtain corresponding outputs $f_1(x)$ and $f_2(x)$, respectively. Then, we compute a loss $\mathcal{L}_c$ to penalize

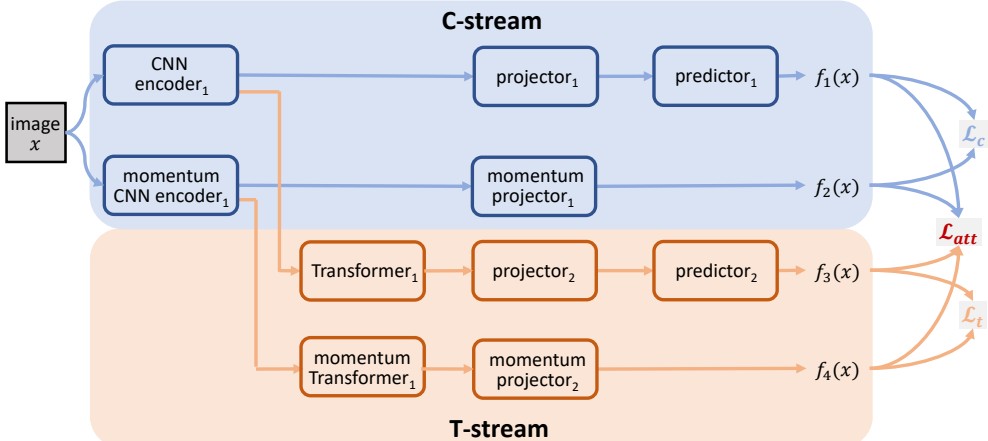

Figure 2: The pipeline of CARE. It consists of C-stream and T-stream. C-stream is similar to the existing SSL framework, and we involve transformers in T-stream. During training, we perform SSL in each stream (i.e., $\mathcal{L}_c$ and $\mathcal{L}_t$), and use T-stream outputs to supervise C-stream (i.e., $\mathcal{L}_{att}$). The CNN encoder becomes attentive via T-stream attention supervision.

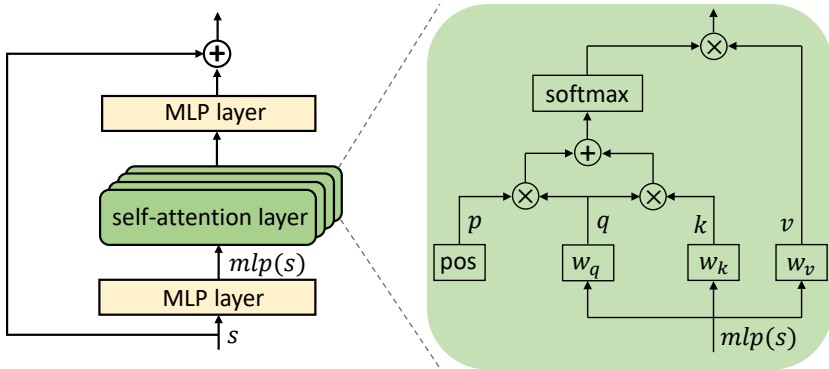

Figure 3: Transformer framework. The architectures of the two transformers in T-stream are the same. Each transformer consists of $n$ attention blocks. We show one attention block on the left, where the detailed structure of one self-attention layer is shown on the right.

the dissimilarity of the outputs. This loss term is the mean squared error of the normalized feature vectors and can be written as what follows:

$$\mathcal{L}_c = 2 - 2 \cdot \frac{\langle f_1(x), f_2(x) \rangle}{\|f_1(x)\|_2 \cdot \|f_2(x)\|_2} \tag{1}$$

where $\|\cdot\|_2$ is the $\ell_2$ normalization, and the $\langle,\rangle$ is the dot product operation. As the inputs of C-stream are from one image, the outputs of C-stream are supposed to become similar during the training process.

### 3.2 Transformer-stream (T-stream)

The T-stream takes the output feature maps of the CNN encoders as its inputs, which are set in parallel to the C-stream. It consists of two transformers, two projectors, and one predictor. The structures of the projectors and the predictor are the same as those in the C-stream. The structures of two transformers share the same architecture, which consists of $n$ consecutive attention blocks where each block contains two Multilayer Perception (MLP) layers with one multi-head self-attention (MHSA) layer in between. We mainly follow the design of [54] to construct the attention block in transformer as shown in Fig. 3. The input feature map (denoted as $s$) of an attention block is first processed by the a MLP layer for dimension mapping, and then passes the self-attention layer and finally the another MLP layer. MHSA consists of multiple attention heads that process the input features in parallel. In

one attention head, as illustrated on the right of Fig. 3, the input feature map is mapped to the query feature ($q$), the key feature ($k$), and the value feature ($k$) via 3 different MLP layers $w_q$, $w_k$, and $w_v$, respectively. As detailed in Eq. (2), the query $q$ and key $k$ are multiplied to form the content-based attention, and $q$ and the position encoding $p$ are multiplied to form the position-based attention.

$$\text{Attention}(q, k, v) = \text{softmax}(\frac{qp^{\text{T}} + qk^{\text{T}}}{\sqrt{d_k}})v \tag{2}$$

where $d_k$ is the dimension of the query and the key. There are learnable parameters in the positional encoding module [46] to output $p$ that follows the shape of $s$. In Eq. (2), we perform matrix multiplication between $q$ and $p^{\text{T}}$, $q$ and $k^{\text{T}}$, and the softmax output and $v$ by treating each pixel as a token [62] (i.e., for a feature map with $c$ channels and spatial dimension of $h \times w$, it forms $h \cdot w$ $c$-dimensional tokens and thus obtains a matrix of size $h \cdot w \times c$). Besides, we perform matrix addition between $qp^{\text{T}}$ and $qk^{\text{T}}$. The output of the second MLP layer is added to the original input feature map $s$ via a residual connection [29], and finally passes a ReLU activation layer.

In T-stream, the outputs of the two transformers are feature maps with 2D dimensions, which are then average-pooled and sent to the projectors and the predictor. We denote the outputs of T-stream as $f_3(x)$ and $f_4(x)$. Following the dissimilarity penalty in Sec. 3.1, we compute $\mathcal{L}_t$ as follows:

$$\mathcal{L}_t = 2 - 2 \cdot \frac{\langle f_3(x), f_4(x) \rangle}{\|f_3(x)\|_2 \cdot \|f_4(x)\|_2}. \tag{3}$$

Besides introducing SSL loss terms in both streams, we use the T-stream output to supervise the C-stream. This attention supervision loss term can be written as:

$$\mathcal{L}_{\text{att}} = \|f_1(x) - f_3(x)\|_2 + \|f_2(x) - f_4(x)\|_2 \tag{4}$$

where the C-stream outputs are required to resemble the T-stream outputs during the training process. Note that the network supervision in Eq. 4 can not be simply considered as the knowledge distillation (KD) process. There are several differences from 3 perspectives: (1) The architecture design between ours and KD is different. In KD [31], a large teacher network is trained to supervise a small student network. In contrast, the CNN backbones are shared by two similar networks in our method. The different modules are only transformers, lightweight projectors, and predictor heads. (2) The training paradigm is different. In KD, the teacher network is typically trained in advance before supervising the student network. In contrast, two branches of our method are trained together from scratch for mutual learning. (3) The loss function in KD is normally the cross-entropy loss while we adopt mean squared error. During KD, supervision losses are also computed between feature map levels. While our method only computes losses based on the network outputs.

### 3.3    Network training

The proposed CARE consists of two streams and the loss terms have been illustrated above. The final objective function for network training can be written as:

$$\mathcal{L}_{\text{total}} = \mathcal{L}_c + \mathcal{L}_t + \lambda \cdot \mathcal{L}_{\text{att}} \tag{5}$$

where $\lambda$ is a constant value controlling the influence of the attention supervision loss. After computing $\mathcal{L}_{\text{total}}$, we perform back-propagation only on the upper branches of the C-stream and T-stream. Specifically in Fig. 2, the CNN encoder$_1$, the projector$_1$, and the predictor$_1$ are updated via the computed gradients in C-stream. Meanwhile, the Transformer$_1$, the projector$_2$, and the predictor$_2$ are updated via computed gradients in T-stream. Afterwards, we perform a moving average update [35, 24, 27] on the momentum CNN encoder$_2$ based on the CNN encoder$_1$, on the momentum projector$_1$ based on the projector$_1$, on the momentum transformer$_1$ based on the transformer$_1$, and on the momentum projector$_2$ based on the projector$_2$. We only use positive samples when training the network. As analyzed in [24], using momentum projectors and a predictor is shown important for self-supervised learning. These modules prevent CNN features from losing generalization abilities during the pretext training. Besides, we experimentally found that the momentum update is effective in preventing trivial solutions. In our network, we adopt a similar design in both streams to facilitate network training and obverse that using only positive samples does not cause model collapse. After pretext training, we only keep the CNN encoder$_1$ in CARE. This encoder is then utilized for downstream recognition scenarios.

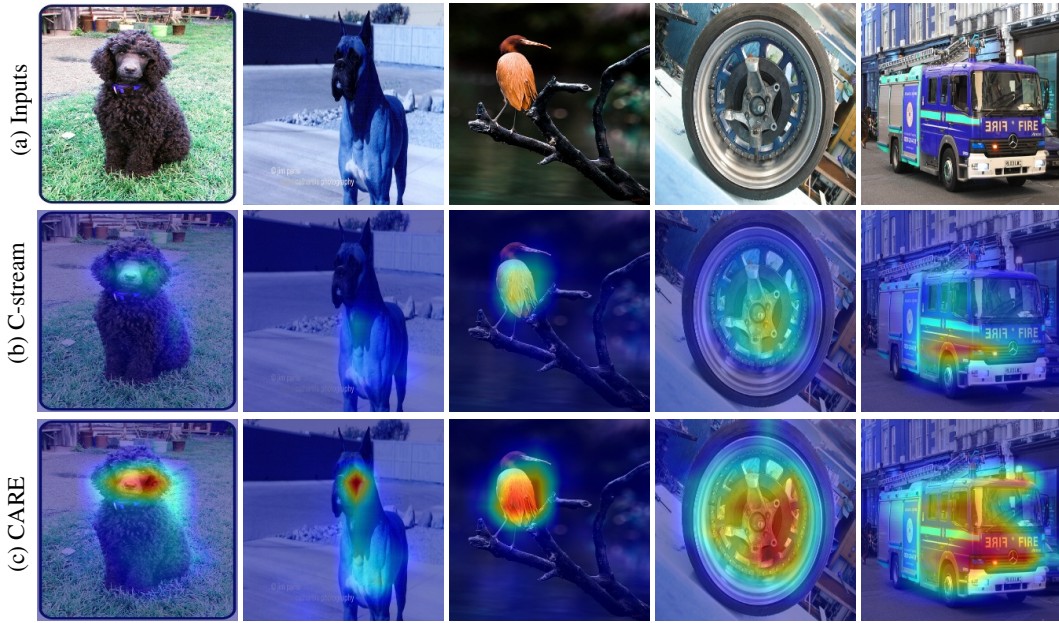

Figure 4: Attention visualization of CNN encoders. We train two ResNet-50 encoders by using only C-stream and the whole CARE method, respectively. By taking the same image in (a) as inputs, the attention maps of these two encoders are shown in (b) and (c). The attention learned via CARE is more intense around the object regions shown in (c). In the attention visualization maps, pixels marked as red indicate the network pays more attention to the current regions.

## 3.4 Visualizations

Our CARE framework improves CNN encoder attention via transformer guidance. We show how encoders attend to the input objects by visualizing their attention maps. The ResNet-50 encoder backbone is used for visualization. We train this encoder for 200 epochs using only C-stream and the whole CARE framework, respectively. For input images, we use [51] to visualize encoder responses. The visualization maps are scaled equally for comparison.

Fig. 4 shows the visualization results. The input images are presented in (a), while the attention maps from the encoders trained with C-stream and CARE are shown in (b) and (c), respectively. Overall, the attention of the encoder trained with CARE is more intense than that with C-stream, which indicates that T-stream in CARE provides effective supervision for CNN encoders to learn to attend to object regions. The T-steam helps CNN encoders adaptively choose to focus on local regions or global regions. For example, when global information is needed for classification, the CNN encoder learned by CARE will pay more attention to the whole object, as in the last column in (c), rather than a limited region, as shown in (b). On the other hand, when local information is sufficient for classification, the CNN encoder learned via CARE will pay more intense attention to the specific regions (e.g., the animals' heads in (c) on the first and second columns). The attention maps shown in the visualization indicate that the CNN encoder becomes attentive on the object region via transformer guidance in our CARE framework.

## 4 Experiments

In this section, we perform experimental validation on our CARE method. First, we introduce implementation details. Then, we compare our CARE method to state-of-the-art SSL methods on standard benchmarks, including image classification, object detection, and semantic segmentation. Furthermore, we conduct ablation studies to analyze each component of the proposed CARE method.

Table 1: Linear evaluations on ImageNet with top-1 accuracy (in %). We highlight the best experimental results under the same model parameters in **bold.**

(a) Classification accuracy by using the ResNet-50 encoder.

| Method | 100ep | 200ep | 400ep |
|---|---|---|---|
| CMC [58] | - | 66.2 | - |
| PCL v2 [34] | - | 67.6 | - |
| SimCLR [12] | 66.5 | 68.3 | 69.8 |
| MoCo v2 [13] | 67.4 | 69.9 | 71.0 |
| SwAV [9] | 66.5 | 69.1 | 70.7 |
| SimSiam [14] | 68.1 | 70.0 | 70.8 |
| InfoMin Aug. [59] | - | 70.1 | - |
| BYOL [24] | 66.5 | 70.6 | 73.2 |
| Barlow Twins [73] | - | - | 72.5 |
| CARE (ours) | **72.0** | **73.8** | **74.7** |

(b) Classification accuracy via CNN and Transformer encoders.

| Method | Arch. | Param. | Epoch | GFlops | Top-1 |
|---|---|---|---|---|---|
| CMC [58] | ResNet-50(2×) | 69M | - | 11.4 | 70.6 |
| BYOL [24] | ResNet-50(2×) | 69M | 100 | 11.4 | 71.9 |
| BYOL [24] | ResNet-101 | 45M | 100 | 7.8 | 72.3 |
| BYOL [24] | ResNet-152 | 60M | 100 | 11.6 | 73.3 |
| BYOL [24] | ViT-S | 22M | 300 | 4.6 | 71.0 |
| BYOL [24] | ViT-B | 86M | 300 | 17.7 | 73.9 |
| MoCo v3 [15] | ViT-S | 22M | 300 | 4.6 | 72.5 |
| CARE (ours) | ResNet-50 | 25M | 200 | 4.1 | **73.8** |
| CARE (ours) | ResNet-50(2×) | 69M | 100 | 11.4 | **73.5** |
| CARE (ours) | ResNet-50(2×) | 69M | 200 | 11.4 | **75.0** |
| CARE (ours) | ResNet-101 | 45M | 100 | 7.8 | **73.5** |
| CARE (ours) | ResNet-152 | 60M | 100 | 11.6 | **74.9** |

## 4.1 Implementation details

**Image pre-processing.** The training images we use during pretext training are from the ImageNet-1k [50] dataset. We follow [24] to augment image data before sending them to the encoders. Specifically, we randomly crop patches from one image and resize them to a fixed resolution of $224 \times 224$. Then, we perform random horizontal flip and random color distortions on these patches. The Gaussian blur, the decolorization, and the solarization operations are also adopted to preprocess these patches.

**Network architectures.** We use ResNet encoder backbones [29] (i.e., ResNet-50, ResNet-101, and ResNet-152) in our experiments. The architectures of the projectors and the predictors are the same and follow [24]. Each projector and predictor consist of a fully-connected layer with a batch normalization and a ReLU [44] activation, followed by another fully-connected layer. The transformer in the T-stream contains $n$ attention blocks as shown in Fig. 3.

**Network training process.** We use the SGD optimizer with a momemtum of 0.9 during pretext training. The base learning rate is set as 0.05 and scaled linearly with respect to the batch size [23] (i.e., $\mathrm{lr}_{\mathrm{base}} = 0.05 \times \mathrm{BatchSize}/256$). We start the pretext training with a warm-up of 10 epochs where the learning rate rises linearly from $10^{-6}$ to the base learning rate ($\mathrm{lr}_{\mathrm{base}}$). Then, we use a cosine decay schedule for the learning rate without restarting it [39, 24] to train the network. The momentum update coefficient of network parameters (denoted as $\tau$) is increased from 0.99 to 1 via a cosine design (i.e., $\tau = 1 - (1 - \tau_{\mathrm{base}}) \cdot (\cos(\pi t/T) + 1)/2$, where $t$ is the current training step and $T$ is the total number of training steps). We train CARE using 8 Tesla V100 GPUs with a batch size of 1024. The automatic mixed precision training strategy [41] is adopted for training speedup.

## 4.2 Comparison to state-of-the-art approaches

We compare feature representations of CNN encoders learned by our method and state-of-the-art SSL methods. Comparisons are conducted on recognition scenarios, including image classification (self-supervised and semi-supervised learning configurations), object detection, and semantic segmentation.

**Self-supervised learning on image classifications.** We follow [27] to use standard linear classification protocol where the parameters of the encoder backbone are fixed and an additional linear classifier is added to the backbone. We train this classifier using SGD for 80 epochs with a learning rate of 0.2, a momentum of 0.9, and a batch size of 256. The ImageNet training set is used for the training and the ImageNet validation set is used for evaluation.

Table 1 shows the linear evaluation results with the top-1 accuracy. We show the classification results by using the ResNet-50 encoder learned via different SSL methods in Table 1a. In this table, our CARE method consistently outperforms other methods under different training epochs. Specifically, our method achieves a 74.7% top-1 accuracy under 400 training epochs, which is 1.5% higher than the second-best method BYOL. Meanwhile, we compare our method to other methods that use CNN

Table 2: Linear evaluations on ImageNet with top-1 and top-5 accuracy (in%). We present the experimental results of different CNN encoders that are trained by using more epochs (e.g., 200 epochs, 400 epochs and 800 epochs).

| Method | Arch. | Param. | Epoch | GFlops | Top-1 | Top-5 |
|---|---|---|---|---|---|---|
| BYOL [24] | ResNet-50 | 25M | 800 | 4.1 | 74.3 | 91.7 |
| BYOL [24] | ResNet-50(2×) | 69M | 800 | 11.4 | 76.2 | 92.8 |
| BYOL [24] | ResNet-101 | 45M | 800 | 7.8 | 76.6 | 93.2 |
| BYOL [24] | ResNet-152 | 60M | 800 | 11.6 | 77.3 | 93.3 |
| CARE (ours) | ResNet-50 | 25M | 200 | 4.1 | 73.8 | 91.5 |
| CARE (ours) | ResNet-50 | 25M | 400 | 4.1 | 74.7 | 92.0 |
| CARE (ours) | ResNet-50 | 25M | 800 | 4.1 | 75.6 | 92.3 |
| CARE (ours) | ResNet-50(2×) | 69M | 200 | 11.4 | 75.0 | 92.2 |
| CARE (ours) | ResNet-50(2×) | 69M | 400 | 11.4 | 76.5 | 93.0 |
| CARE (ours) | ResNet-50(2×) | 69M | 800 | 11.4 | 77.0 | 93.2 |
| CARE (ours) | ResNet-101 | 45M | 200 | 7.8 | 75.9 | 92.7 |
| CARE (ours) | ResNet-101 | 45M | 400 | 7.8 | 76.9 | 93.3 |
| CARE (ours) | ResNet-101 | 45M | 800 | 7.8 | 77.2 | 93.5 |
| CARE (ours) | ResNet-152 | 60M | 200 | 11.6 | 76.6 | 93.1 |
| CARE (ours) | ResNet-152 | 60M | 400 | 11.6 | 77.4 | 93.6 |
| CARE (ours) | ResNet-152 | 60M | 800 | 11.6 | 78.1 | 93.8 |

Table 3: Image classification by using semi-supervised training on ImageNet with Top-1 and Top-5 accuracy (in %). We report our method with more training epochs in the supplementary files.

(a) Classification accuracy by using the ResNet-50 encoder.

| Method | Epoch | Top-1 | | Top-5 | |
|---|---|---|---|---|---|
| | | 1% | 10% | 1% | 10% |
| Supervised [75] | - | 25.4 | 56.4 | 48.4 | 80.4 |
| PIRL [43] | - | - | - | 57.2 | 83.8 |
| SimCLR [12] | 800 | 48.3 | 65.6 | 75.5 | 87.8 |
| BYOL [24] | 800 | 53.2 | 68.8 | 78.4 | 89.0 |
| CARE (Ours) | 400 | **60.0** | **69.6** | **81.3** | **89.3** |

(b) Classification accuracy by using other CNN encoders.

| Method | Arch. | Epoch | Top-1 | | Top-5 | |
|---|---|---|---|---|---|---|
| | | | 1% | 10% | 1% | 10% |
| BYOL [24] | ResNet-50(2×) | 100 | 55.6 | 66.7 | 77.5 | 87.7 |
| BYOL [24] | ResNet-101 | 100 | 55.8 | 65.8 | 79.5 | 87.4 |
| BYOL [24] | ResNet-152 | 100 | 56.8 | 67.2 | 79.3 | 88.1 |
| CARE (ours) | ResNet-50(2×) | 100 | 57.4 | 67.5 | 79.8 | 88.3 |
| CARE (ours) | ResNet-50(2×) | 200 | 61.2 | 69.6 | 82.3 | 89.5 |
| CARE (ours) | ResNet-101 | 100 | 57.1 | 67.1 | 80.8 | 88.2 |
| CARE (ours) | ResNet-101 | 200 | 62.2 | 70.4 | 85.0 | 89.8 |
| CARE (ours) | ResNet-152 | 100 | 59.4 | 69.0 | 82.3 | 89.0 |

and transformer (i.e., ViT [19]) encoders in Table 1b. The results show that, under similar number of parameters of CNN and transformer encoders (i.e., ResNet-50 and ViT-S), CARE achieves higher accuracy than other SSL methods. This indicates that CARE improves CNN encoders to outperform transformer encoders by utilizing visual attention. Besides, we provide the linear classification results of different CNN encoders (e.g., ResNet-101 and ResNet-152) with more training time in Table 2, where our CARE method also consistently prevails.

**Semi-supervised learning on image classifications.** We evaluate our CARE method by using a semi-supervised training configuration on the ImageNet dataset. After pretext training, we finetune the encoder by using a small subset of ImageNet's training set. We follow the semi-supervised learning protocol [24, 12] to use 1% and 10% training data (the same data splits as in [12]) during finetuning. Table 3 shows the top-1 and top-5 accuracy on the ImageNet validation set. The results indicate that our CARE method achieves higher classification accuracy than other SSL methods under different encoder backbones and training epochs.

**Transfer learning to object detection and semantic segmentation.** We evaluate CARE's representations on the downstream object detection and semantic segmentation scenarios. We use the standard VOC-07, VOC-12, and COCO datasets [20, 37]. We follow the standard protocol [27] to integrate the pretext trained CNN encoder into Faster-RCNN [48] when evaluating object detection results on VOC-07 and VOC-12 datasets. On the other hand, we integrate this encoder into Mask-RCNN [28] when evaluating object detection and semantic segmentation in COCO dataset. The ResNet-50 encoder is used in all the methods. All detectors are finetuned for 24k iterations using VOC-07 and

Table 4: Transfer learning to object detection and instance segmentation. The best two results in each column are in bold. Our method achieves favorable detection and segmentation performance by using limited training epochs.

| Method | Epoch | COCO det. | | | COCO instance seg. | | | VOC07+12 det. | | |
|---|---|---|---|---|---|---|---|---|---|---|
| | | $AP^{bb}$ | $AP^{bb}_{50}$ | $AP^{bb}_{75}$ | $AP^{mk}$ | $AP^{mk}_{50}$ | $AP^{mk}_{75}$ | AP | $AP_{50}$ | $AP_{75}$ |
| Rand Init | - | 26.4 | 44.0 | 27.8 | 29.3 | 46.9 | 30.8 | 33.8 | 60.2 | 33.1 |
| Supervised | 90 | 38.2 | 58.2 | 41.2 | 33.3 | 54.7 | 35.2 | 53.5 | 81.3 | 58.8 |
| PIRL[43] | 200 | 37.4 | 56.5 | 40.2 | 32.7 | 53.4 | 34.7 | 55.5 | 81.0 | 61.3 |
| MoCo[27] | 200 | 38.5 | 58.3 | 41.6 | 33.6 | 54.8 | 35.6 | 55.9 | 81.5 | 62.6 |
| MoCo-v2[13] | 200 | 38.9 | 58.4 | 42.0 | 34.2 | 55.2 | 36.5 | 57.0 | 82.4 | 63.6 |
| MoCo-v2[13] | 800 | 39.3 | 58.9 | 42.5 | 34.4 | 55.8 | 36.5 | 57.4 | 82.5 | 64.0 |
| SwAV[9] | 200 | 32.9 | 54.3 | 34.5 | 29.5 | 50.4 | 30.4 | - | - | - |
| SwAV[9] | 800 | 38.4 | 58.6 | 41.3 | 33.8 | 55.2 | 35.9 | 56.1 | 82.6 | 62.7 |
| Barlow Twins[73] | 1000 | 39.2 | 59.0 | 42.5 | 34.3 | 56.0 | 36.5 | 56.8 | 82.6 | 63.4 |
| BYOL [24] | 200 | 39.2 | 58.9 | 42.4 | 34.3 | 55.6 | 36.7 | 57.0 | 82.3 | 63.6 |
| CARE (Ours) | 200 | **39.4** | **59.2** | **42.6** | **34.6** | **56.1** | **36.8** | **57.7** | **83.0** | **64.5** |
| CARE (Ours) | 400 | **39.6** | **59.4** | **42.9** | **34.7** | **56.1** | **36.9** | **57.9** | **83.0** | **64.7** |

Table 5: Transfer learning to object detection and instance segmentation with the Mask R-CNN R50-FPN detector. The best two results in each column are in bold. Our method achieves favorable detection and segmentation performance by using limited training epochs.

| Method | Epoch | COCO det. | | | COCO instance seg. | | |
|---|---|---|---|---|---|---|---|
| | | $AP^{bb}$ | $AP^{bb}_{50}$ | $AP^{bb}_{75}$ | $AP^{mk}$ | $AP^{mk}_{50}$ | $AP^{mk}_{75}$ |
| Rand Init | - | 31.0 | 49.5 | 33.2 | 28.5 | 46.8 | 30.4 |
| Supervised | 90 | 38.9 | 59.6 | 42.7 | 35.4 | 56.5 | 38.1 |
| PIRL[43] | 200 | 37.5 | 57.6 | 41.0 | 34.0 | 54.6 | 36.2 |
| MoCo[27] | 200 | 38.5 | 58.9 | 42.0 | 35.1 | 55.9 | 37.7 |
| MoCo-v2[13] | 200 | 38.9 | 59.4 | 42.4 | 35.5 | 56.5 | 38.1 |
| MoCo-v2[13] | 800 | 39.4 | 59.9 | 43.0 | 35.8 | 56.9 | 38.4 |
| SwAV[9] | 200 | 38.5 | **60.4** | 41.4 | 35.4 | 57.0 | 37.7 |
| BYOL [24] | 200 | 39.1 | 59.5 | 42.7 | 35.6 | 56.5 | 38.2 |
| BYOL [24] | 400 | 39.2 | 59.6 | 42.9 | 35.6 | 56.7 | 38.2 |
| BYOL [24] | 800 | 39.4 | 59.9 | 43.0 | 35.8 | 56.8 | **38.5** |
| Barlow Twins[73] | 1000 | 36.9 | 58.5 | 39.7 | 34.3 | 55.4 | 36.5 |
| CARE (Ours) | 200 | **39.5** | 60.2 | **43.1** | **35.9** | **57.2** | **38.5** |
| CARE (Ours) | 400 | **39.8** | **60.5** | **43.5** | **36.2** | **57.4** | **38.8** |

VOC-12 training sets and are evaluated on the VOC-07 test set. On the COCO dataset, all models are finetuned via the $1\times$ schedule. The results are averaged over five independent trials.

Table 4 shows the evaluation results. Compared to the supervised method, our CARE improves detection (i.e., $1.4\%$ on COCO and $4.4\%$ on VOC) and segmentation (i.e., $1.4\%$ on COCO) performance. On the other hand, our CARE method compares favorably against other SSL methods. Note that the results of our method are reported under 200 and 400 epochs, which are still higher than other methods under 800 epochs. This indicates the effectiveness of our CARE method on learning the CNN encoder backbone. The comparisons on COCO datasets are similar to those on VOC. Specifically, our CARE method achieves a $0.5\%$ $AP^{bb}$ increase and a $0.4\%$ $AP^{mk}$ increase upon MoCo v2 under 200 epochs. The performance improvement is mainly brought by the visual attention integration from transformer guidance.

Besides, we further evaluate CARE's representation on the COCO dataset via a more powerful detector, the feature pyramid network (FPN) [36], and report the results in Table 5. We follow the same evaluation protocol introduced above. The detectors are trained with $1\times$ schedule (90k iterations) for fair comparisons. Again, CARE trained for 200/400 epochs outperforms other state-of-the-art SSL methods trained for 800/1000 epochs on object detection and semantic segmentation on the COCO dataset, which suggests that the CNN encoder in CARE is empowered by the attention mechanism of the transformer which supervises the CNN encoder in the pretraining.

| Table 6: Analysis on $\lambda$. | | | Table 7: Analysis on $n$. | | | Table 8: Analysis on the positional encoding. | |
|---|---|---|---|---|---|---|---|
| $\lambda$ | Top-1 | | $n$ | Top-1 | | Position encoding | Top-1 |
| 0 | 70.52 | | 2 | 71.08 | | none | 69.49 |
| 1 | 70.88 | | 3 | 71.60 | | sin-cos absolute [62] | 66.68 |
| 10 | 70.96 | | 4 | 71.79 | | learnable absolute [55] | 72.01 |
| 100 | 72.06 | | 5 | 72.06 | | learnable relative [52] | 72.06 |
| 250 | 72.00 | | 6 | 69.37 | | | |

### 4.3 Ablation studies

In our CARE method, visual attention is explored via transformers to supervise C-stream. We analyze the influence of attention supervision by using different values of $\lambda$ in Eq. (5). Also, we analyze how the number of attention blocks and the positional encoding affect feature representations. Besides, we further take an investigation of the sequential and parallel design of T-stream. We use the ResNet-50 as encoder backbone and the number of training epochs is set to 100. The top-1 image classification accuracy on ImageNet via SSL is reported to indicate feature representation effectiveness.

**Supervision influence** $\lambda$. We study how the attention supervision term $\mathcal{L}_{\mathrm{att}}$ in Eq. (5) affects feature representation by using different values of $\lambda$. Table 6 shows the evaluation results. When we set $\lambda$ as 0, the CNN encoder is learned without attention supervision. In this case, the performance decreases significantly. When we increase the value of $\lambda$, the attention supervision increases as well. We observe that $\lambda = 100$ achieves the best performance and adopt this setting in the overall experiments.

**Number of attention blocks.** We analyze how the capacity of transformer in T-stream affects the recognition performance. We set $n = [2, ..., 6]$ to constitute five transformers in T-stream with increasing capacities. Then, we report the recognition performance of the corresponding CNN encoders in Table 7. When $n$ is larger, the transformer capacity increases and stronger visual attention are explored. This attention supervises C-stream to improve the encoder. However, the recognition performance drop when $n = 6$. This may be due to the broken balance between the attention loss $\mathcal{L}_{\mathrm{att}}$ and the original SSL loss $\mathcal{L}_c$. In our experiment, we set $n = 5$ in our CARE method.

**Positional encoding.** We analyze how positional encoding affects the final performance. Numerous positional encoding settings [62, 55, 52] are adopted for analysis in Table 8. Without positional encoding, the performance decreases significantly. Meanwhile, the conventional fixed sine-cosine encoding setting is not suitable for CARE. We experimentally find that using learnable parameter configuration in positional encoding improves recognition performance and adopt [52] in CARE.

**Sequential design** *v.s.* **parallel design.** Experimental results on training ResNet-50 with 100 epochs indicate that parallel design is effective than sequential design (i.e., 72.02% v.s. 69.32%). This is because during the sequential training process, both the CNN encoders and the transformers are optimized together rather than the CNN encoders themselves. This prevents attention supervision from training the CNN encoders thoroughly.

## 5 Concluding remarks

Transformers have advanced visual recognition via attention exploration architectures. In self-supervised visual representation learning, studies emerge to utilize transformer backbones for recognition performance improvement. This indicates that visual attention explored by the transformers benefit SSL methods. Motivated by this success, we investigate how to explore visual attention effectively to benefit CNN encoders in SSL. We propose CARE to develop a parallel transformer stream together with the CNN stream. The visual attention is thus explored via transformers to supervise the CNN stream during SSL. Although the limitation occurs that more computational cost is spent on the SSL and attention supervision loss term, the learned CNN encoder becomes attentive with transformer guidance and does not consume more costs in downstream tasks. Experiments on the standard visual recognition benchmarks, including image classification, object detection, and semantic segmentation, indicate that CARE improves CNN encoder backbones to a new state-of-the-art performance.

**Acknowledgement**. This work is supported by CCF-Tencent Open Fund, the General Research Fund of Hong Kong No.27208720 and the EPSRC Programme Grant Visual AI EP/T028572/1.

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
