# OpenReview forum: "Revitalizing CNN Attention via Transformers in Self-Supervised Visual Representation Learning"
_NeurIPS.cc/2021/Conference — NeurIPS 2021 Poster_

### Official Review · Reviewer_hShL · 2021-07-07

**Rating:** 8
**Confidence:** 4

**Summary:**

In this paper, a self-supervised visual representation learning approach is developed to improve CNN backbone encoders. The transformers are introduced to improve visual attention abilities of transformers. The evaluation on the experiments has shown the proposed approach achieves favorable performance.

**Ethics Review Area:**

["I don’t know"]

**Limitations And Societal Impact:**

The last two branches still involve projectors and a predictor. This seems not identical for the whole pipeline. These two branches are set to improve the visual attentions of the backbone encoder output. Using transformer seems sufficient to fulfill this task. Is there any possibility to remove these modules to reduce pipeline capacity?

In the experiments, some backbone encoders are evaluated by training 100 epochs. Using more training time shall be helpful to validate the performance improvement upon downstream scenarios.

On the last paragraph, societal impact is discussed.

-----------------------------Post Rebuttal---------------------------
My concerns have been well addressed in the response. After reading reviews from other reviewers, I am sure this is a good paper and increase my score to 8.

**Main Review:**

The main novelty of this work is to develop a four branches paradigm for self-supervised visual representation learning. The first two branches are similar to prior SSL works where there are backbone encoders, projectors, and the predictor. The last two branches are involved with transformers. And these branches are updated via either gradient back-propagations or a momentum schedule. The last two branches are new and shown to improve the backbone encoders.

The overall presentation seems complete and the technical components are clear. The paper organization is straightforward, and the visualizations are useful to show the attentions. The implementation code is provided in the supplementary files.

Results on the downstream scenarios (image classification, semi-supervised classification, object detection, and semantic segmentation) show that there is an improvement upon existing SSL methods.

**Time Spent Reviewing:**

3

---

> ### Author Response · Authors · 2021-08-10
> **Response to Reviewer hShL: Discussions on the T-stream architectures and the extended experiments.**
>
> We thank the reviewer for the comments and we answer the raised questions below. All discussions and results will be added to the revised manuscript.
>
> ***1. Removal of the projectors and predictor in T-stream.***
>
> Using projectors and predictor in T-stream is important during training. The importance of these modules has been analyzed in [24] to improve feature generalizations as well as stabilize network training. As we conduct self-supervised learning on C-stream and T-stream separately, we need the projectors and predictor in each stream during the training process.  The experimental results on training ResNet-50 with 100 epochs also validate the effectiveness of our parallel design (70.02% acc without the projectors and predictor in T-stream v.s. 72.02% with the projectors and predictor in T-stream).
>
> ***2. Evaluations by using more training time.***
>
> We have conducted the linear evaluation experiments on different CNN backbones trained with more epochs. The results are shown below. We here choose a state-of-the-art SSL framework, BYOL, for an intuitive comparison.
>
> |  Method   | Arch.  | Param | Epoch | Top-1 | Top-5 |
> | :-----| :----| :----: |:-----:| ----: |----: |
> | BYOL  | ResNet-50  |  25M  | 800  | 74.3  | 91.7 |
> |  BYOL  | ResNet-50(2x)  |  69M  | 800  | 76.2  | 92.8 |
> |  BYOL  | ResNet-101 |  45M  | 800  | 76.6  | 93.2 |
> |  BYOL  | ResNet-152  |  60M  | 800  | 77.3  | 93.2 |
> |  CARE  | ResNet-50  |  25M  | 400  | 74.7  | 92.0 |
> |  CARE  | ResNet-50  |  25M  | 800  |  75.6  | 92.3  |
> |  CARE  | ResNet-50(2x)  |  69M  | 400  |  76.5  | 93.0  |
> |  CARE  | ResNet-50(2x)  |  69M  | 800  |  77.1  | 93.2  |
> |  CARE  | ResNet-101 |  45M  | 200  |  75.9  | 92.7  |
> |  CARE  | ResNet-101 |  45M  | 400  |  76.9  | 93.3  |
> |  CARE  | ResNet-101 |  45M  | 800  |  77.3  | 93.5  |
> |  CARE  | ResNet-152  |  60M  | 200  |  76.6 | 93.1  |
> |  CARE  | ResNet-152  |  60M  | 400  | 77.4  | 93.6  |
> |  CARE  | ResNet-152  |  60M  | 800  |  78.1  | 93.8  |
>
> Our method performs favorably against the state-of-the-art methods under different training epochs.

---

### Official Review · Reviewer_Cw1s · 2021-07-15

**Rating:** 7
**Confidence:** 4

**Summary:**

This paper uses transformers to improve visual representation learning. The two view augmentation and the response generation solution from existing approaches are boosted by transformers for visual attention enhancement. The main contribution lies in a structure with two transformer branches to improve CNN attention and supervise the original CNN branches. I can see that the attention mechanism obtains gains due to the parallel design. The proposed method shows superiority, compared to existing work.

**Limitations And Societal Impact:**

They were mentioned in the paper.

**Main Review:**

I see some merits, which include:
1. The proposed research is credible, the objectives are comprehensive, clearly specified, and of high scientific quality. The paper clearly identifies a number of highly relevant challenges and proposes an appropriate methodology to address them. In the supplementary files, the code is provided as well.
2. The proposed transformer branch looks interesting because it advances the original SSL framework, where only CNN branches are introduced. On top of it, the feature response from the CNN branches is integrated with transformer attentions.
3. The proposed method is evaluated on several benchmark datasets for different vision tasks, for example, image classification, object detection, and semantic segmentation scenarios. I would say the achieved results are solid.

However, some questions are:
1. The proposed T-stream includes a (momentum) transformer, a (momentum) projector, and a predictor. It looks quite complicated. Why is momentum strategy utilized in these branches? How does it work? What are the benefits of using it? All these things are not clear in the paper.
2. I wonder what would happen if you enable a sequential way, e.g., CNN+Transform? Why does (not) it work?
3. It seems contradictory - on the one hand, the paper claims that the transformer improves CNNs' attention; on the other hand, I saw that there are convolution layers within the transformer structure. Why is that?

**Time Spent Reviewing:**

2

---

> ### Author Response · Authors · 2021-08-10
> **Response to Reviewer Cw1s: Discussions on the momentum configurations, sequential design and the convolution layers.**
>
> We thank the reviewer for the comments and we answer the raised questions below. All discussions and results will be added to the revised manuscript.
>
> ***1. Momentum configurations.***
>
> In [24], using momentum projectors and a predictor is shown important for self-supervised learning. These modules prevent CNN features from losing generalization abilities during the pretext training, which are unsuitable when adapting to downstream scenarios. Meanwhile, using these modules is effective in preventing trivial solutions during training. We follow this design in our T-stream to add (momentum) transformer, a (momentum) projector, and a predictor. Meanwhile, we update the network with a momentum update operation following [24, 27]. This momentum update is also effective in preventing trivial solutions. The detailed momentum update is shown in line 023 of Algorithm 1 in the supplementary materials, where the parameters of the momentum transformer are computed by a weighted average of its own parameters and those from the transformer.
>
> ***2. Sequential design.***
>
> Using a sequential design to connect a CNN encoder and a transformer does not improve encoder performance. This is because the encoder and the transformer are regarded as a united network rather than two modules. During self-supervised training, this united network is optimized instead of the CNN encoder. As a result, the features from the CNN encoder are not attentive after training. In contrast, the parallel design uses the attentive features to supervise the CNN branch. The encoder is trained to become attentive after training (as shown in Figure 4 in our manuscript). Experimental results on training ResNet-50 with 100 epochs indicate that parallel design is effective than sequential design (i.e., 72.02% v.s. 69.32%).
>
> ***3. Convolution layers within transformers.***
>
> This is perhaps a misunderstanding. In the transformer, we use 1x1 conv layers for the dimension mapping purpose. The conv 1x1 layer (with kernel size 1 and stride 1) within the transformer actually acts like a fully-connected layer. The input tokens from feature maps are mapped via the fully-connected layer to the query (Q), key (K), and value (V) for scaled dot-product operation [58]. Thus, from the implementation aspects, the transformers consist of attention blocks and fully-connected layers, which are implemented by 1x1 conv layers. We will revise the conv layer to fully-connected layer in the figure.

---

### Official Review · Reviewer_XzkT · 2021-07-16

**Rating:** 8
**Confidence:** 4

**Summary:**

A self-supervised visual representation learning method is proposed in this paper. Besides the prevalent two-branch CNN architecture design, the two-branch transformer structure is introduced to improve CNN backbone features via attention enhancement. The proposed method is evaluated on the several visual recognition benchmarks and shown effective through extensive experiments.

**Limitations And Societal Impact:**

The proposed method uses T-stream output to supervise C-stream. The network supervision shares some similarities to knowledge distillation where the teacher network is used to supervise the student network. A discussion upon knowledge distillation is useful to understand the contribution better. The negative societal impact has been illustrated and addressed on the last paragraph and this reviewer thinks it to be adequate

**Main Review:**

The proposed method, named  CARE, introduces a transformer to improve the CNN backbones during SSL. This is novel and interesting. Prevalent visual SSL methods (e.g., BYOL, SimCLR) use two CNN branches to create augmented two views of one image, which is formulated as  C-stream in CARE. Differently, CARE introduces two branches containing transformers to improve CNN backbone attentions, which is named as T-stream. The improvement is fulfilled via network supervision from T-stream to C-stream during training. The related work is adequately cited.
The proposed method sounds reasonable. The learned attention within the CNN backbones is visualized to show the training effectiveness. The overall framework seems complete. In the experiments, thorough evaluations are conducted to evaluate the proposed method including image classification (linear probe and semi-supervised), object detection, and semantic segmentation.
The overall pipeline is presented well and the technical details are not difficult to follow. The figures of network structure comparison, pipeline illustration, and visualizations are helpful to understand the proposed pipeline. Both the pseudo code and the source code are provided and the proposed method is trustworthy.
The overall experimental results seem promising and clearly improves the sota performance. Using two additional branches to improve the original two branches is new and motivates potential further development. It provides new thinking of using the advantage of one type of network to improve another type of network. This is a unique experimental approach to the visual SSL field.

------------------------------------------------Post Rebuttal--------------------------------------------------------------------------

The responses have addressed the issues listed in the previous comment, so I will keep my rating.

**Time Spent Reviewing:**

4

---

> ### Author Response · Authors · 2021-08-10
> **Response to Reviewer XzkT: Discussions on the knowledge distillation.**
>
> We thank the reviewer for the comments and we respond to the suggestions below. All discussions and results will be added to the revised manuscript.
>
> The proposed method aims at transferring the attention ability from Transformers to CNNs, while knowledge distillation (KD) transfers the prediction of a teacher network to a student network. There are several differences as illustrated below.
>
> 1. **The architecture design** between ours and KD is different. In KD [a], a large teacher network is trained to supervise a small student network. In contrast, the CNN backbones are shared by two similar networks in our method. The different modules are only transformers, lightweight projectors, and predictor heads.
>
> 2. **The training paradigm** is different. In KD, the teacher network is typically trained in advance before supervising the student network. In contrast, two branches of our method are trained together from scratch for mutual learning.
>
> 3. **The loss function** in KD is normally the cross-entropy loss while we adopt mean squared error. During KD, supervision losses are also computed between feature map levels. While our method only computes losses based on the network outputs.
>
> We will add these discussions to the revised manuscript.
>
> [a]. Geoffrey Hinton, Oriol Vinyals, and Jeff Dean.  Distilling the knowledge in a neural network. In *arxiv, preprint arXiv:1503.02531*, 2015.

---

### Official Review · Reviewer_h77N · 2021-07-19

**Rating:** 7
**Confidence:** 4

**Summary:**

Transformers are utilized for representation learning. Instead of using them as backbones
to be optimized, this paper proposes to improve CNN backbone attentions via transformers
in the form of network prediction supervision. In the experimental validation, the proposed
method CARE is shown to improve the original representation learning methods with only
ConvNets introduced at the 100, 200, 400 epoches.

**Limitations And Societal Impact:**

See above. The societal impact is shown on the last page of the manuscript.

**Main Review:**

In this paper, a new self-supervised learning approach is designed to improve CNN
backbones. It shares similarities with sota pipeline in that images are created via two views
and measured representations in different branches. The main difference resides on the
additional branches introduced where there are transformers. The visual attentions of
backbone features are enhanced via transformers. Then, these features are used to
supervise original representation learning branches. CNN encoders are thus learned to
become attentive, which is guided by the transformer.

As a representation learning approach, the proposed method provides a potential to
improve backbone networks with transformers. It differs from contemporary representation
learning methods that focus on learning transformer backbones. This is a novel direction
for the representation learning pipeline where additional network properties (i.e., visual
attention from transformer) are introduced to gain the target backbones.

The transformer is updated by a moving average, which is similar to that of predictors in
CNN branches. Why using this move average instead of gradients back-propagation?
Furthermore, the transformer is designed in parallel rather than in sequence. Any
motivation on how to make this design? A direct explanation will improve the motivation of
the proposed method.

In the linear evaluation of the experiments, the performance of using ResNet-50 is reported
at 400 epochs while other backbones are not (e.g., ResNet-101 at 100 epochs and
ResNet-152 at 100 epochs). A consistent training by using same epochs will make the
results more convincing and the comparison more thorough.

------------------------------------------------Post Rebuttal--------------------------------------------------------------------------

My questions are well-addressed. After reading the opinions and reviews from other reviewers, I will increase the final score to 7.


**Time Spent Reviewing:**

5 hours

---

> ### Author Response · Authors · 2021-08-10
> **Response to Reviewer h77N: Discussions on moving average strategy, parallel design, and the extended experiments.**
>
> We thank the reviewer for the comments and we answer the raised questions below.
>
> ***1. Using moving average instead of gradient back-propagations.***
>
> The moving average update of the momentum transformer is effective in preventing the CNN encoder from converging to trivial solutions. An analysis is shown in [14] where the moving average updating strategy of momentum modules (i.e., momentum projector and predictor) stabilizes the training process. Similar approaches have been adopted in [24, 27], and we utilize this strategy to update the momentum transformer.
>
> ***2. Parallel transformer.***
>
> We set the transformers in parallel to the C-stream. This design utilizes the enhanced visual attention to supervise the CNN branch. Thus, the CNN encoder is guided to produce attentive features in training (as shown in Figure 4 in our manuscript). In contrast, we find that a sequential design does not improve CNN feature attentions. During the sequential training process, both the CNN encoder and the transformer are optimized together rather than the CNN encoder itself. This prevents attention supervision from training the CNN encoder thoroughly. The experimental results indicate that when training ResNet-50 with 100 epochs, the parallel design achieves 72.02% accuracy, while the sequential design achieves 69.32%.
>
> ***3. Consistent results under the same training epochs.***
>
> As suggested, we provide more thorough linear evaluation results compared to a state-of-the-art SSL method (i.e., BYOL) using different CNN backbones. The comparisons show that the proposed method performs favorably against BYOL. We will add these results to the revised manuscript.
>
> |  Method   | Arch.  | Param | Epoch | Top-1 | Top-5 |
> | :-----| :----| :----: |:-----:| ----: |----: |
> | BYOL  | ResNet-50  |  25M  | 800  | 74.3  | 91.7 |
> | BYOL  | ResNet-50(2x)  |  69M  | 800  | 76.2  | 92.8 |
> | BYOL  | ResNet-101 |  45M  | 800  | 76.6  | 93.2 |
> | BYOL  | ResNet-152  |  60M  | 800  | 77.3  | 93.3 |
> |  CARE  | ResNet-50  |  25M  | 400  | 74.7  | 92.0 |
> |  CARE  | ResNet-50  |  25M  | 800  |  75.6  | 92.3  |
> |  CARE  | ResNet-50(2x)  |  69M  | 400  |  76.5  | 93.0  |
> |  CARE  | ResNet-50(2x)  |  69M  | 800  |  77.1  | 93.2  |
> |  CARE  | ResNet-101 |  45M  | 200  |  75.9  | 92.7  |
> |  CARE  | ResNet-101 |  45M  | 400  |  76.9  | 93.3  |
> |  CARE  | ResNet-101 |  45M  | 800  |  77.3  | 93.5  |
> |  CARE  | ResNet-152  |  60M  | 200  |  76.6 | 93.1  |
> |  CARE  | ResNet-152  |  60M  | 400  | 77.4  | 93.6  |
> |  CARE  | ResNet-152  |  60M  | 800  |  78.1  | 93.8  |

---

### Author Response · Authors · 2021-08-10
**General Response**

We sincerely appreciate all reviewers' efforts in reviewing our paper and giving insightful comments and valuable suggestions. We are glad to find that the reviewers generally acknowledge the following novelty and contributions of our work.

- **Training Framework.** Unlike the contemporary representation learning methods that focus on learning either CNN or Transformer backbones alone, this work investigates how to effectively explore visual attention from Transformer to benefit CNN encoders [h77N, XzkT, Cw1s, hShL]. We hope our work will inspire re-thinking on using the advantages of one type of network to benefit another one in SSL tasks.

- **Experiments.**  Thorough evaluations are conducted on the downstream tasks, including image classification (linear evaluation and semi-supervised classification), object detection, and semantic segmentation. The effectiveness of the proposed method has been validated [XzkT, Cw1s, hShL].

As suggested by the reviewers, we would like to include the following contents in our revised manuscript to further improve our paper. We summarize the major revision as follows. Our detailed responses can be found in the following response sections to the reviewers.

- **Extended experiments.**  Linear evaluation experiments on different CNN backbones trained with more epochs (e.g., 200 epochs, 400 epochs, and 800 epochs) will be added [h77N, hShL].

- **Discussion on knowledge distillation.** For a better understanding, we will make detailed explanations on the similarities and differences between traditional knowledge distillation and our proposed CARE method [XzkT].

- **Explanations on the moving average (momentum) strategy.** We will make further explanations on the relationship between the designed moving average strategy and the collapse problems [h77N, Cw1s] .

- **Explanations on the T-stream architectures design.** The motivation behind the careful design of T-stream architectures (e.g., the projectors, predictors, and the conv layers in Transformer blocks) will be elaborated [Cw1s, hShL].

---

### Decision · Program_Chairs · 2021-09-27

**Decision:**

Accept (Poster)

**Comment:**

The rebuttal addressed all of the reviewers concerns, and all reviewers recommend acceptance. The AC agrees with this recommendation.